# Impact of Different Lignin Sources on Nitrogen−Doped Porous Carbon toward the Electrocatalytic Oxygen Reduction Reaction

**DOI:** 10.3390/ijerph20054383

**Published:** 2023-03-01

**Authors:** Zheng Li, Yuwei Feng, Xia Qu, Yantao Yang, Lili Dong, Tingzhou Lei, Suxia Ren

**Affiliations:** 1Institute of Urban & Rural Mining, Changzhou University, Changzhou 213164, China; 2Changzhou Key Laboratory of Biomass Green, Safe & High Value Utilization Technology, Changzhou 213164, China

**Keywords:** lignin, carbon material, nitrogen doping, oxygen reduction reaction

## Abstract

Lignin is an ideal carbon source material, and lignin−based carbon materials have been widely used in electrochemical energy storage, catalysis, and other fields. To investigate the effects of different lignin sources on the performance of electrocatalytic oxygen reduction, different lignin−based nitrogen−doped porous carbon catalysts were prepared using enzymolytic lignin (EL), alkaline lignin (AL) and dealkaline lignin (DL) as carbon sources and melamine as a nitrogen source. The surface functional groups and thermal degradation properties of the three lignin samples were characterized, and the specific surface area, pore distribution, crystal structure, defect degree, N content, and configuration of the prepared carbon−based catalysts were also analyzed. The electrocatalytic results showed that the electrocatalytic oxygen reduction performance of the three lignin−based carbon catalysts was different, and the catalytic performance of N−DLC was poor, while the electrocatalytic performance of N−ELC was similar to that of N−ALC, both of which were excellent. The half−wave potential (E_1/2_) of N−ELC was 0.82 V, reaching more than 95% of the catalytic performance of commercial Pt/C (E_1/2_ = 0.86 V) and proving that EL can be used as an excellent carbon−based electrocatalyst material, similar to AL.

## 1. Introduction

To solve the increasingly serious energy crisis and reduce environmental pollution, researchers have focused on developing new energy sources. Among them, fuel cells have attracted much attention due to their advantages, such as rich fuel sources, strong energy conversion ability, low operating temperature, and low environmental pollution [1,2,3]. The oxygen reduction reaction (ORR) is an important cathode reaction in fuel cells and metal–air batteries [4,5]. However, due to the slow kinetics of the ORR, which restricts the energy conversion, electrocatalysts are needed to accelerate the reaction kinetics. To date, platinum−based catalysts are still the most effective ORR catalysts. However, due to their low reserves, high cost, and poor cycle stability, platinum−based catalysts have difficulty meeting the needs of large−scale applications [6]. Therefore, the search for nonmetallic catalysts with low cost and excellent catalytic performance has become a research hotspot.

Porous carbon materials with good electrical conductivity, stable chemical properties, high specific surface area, and abundant porous structure are widely used in electrochemical catalytic research. Lignin is the second−largest biomass resource, second only to cellulose [7,8,9]. Lignin is a byproduct of industri processes that mainly include pulp− and papermaking (i.e., alkaline lignin) and biorefining waste residues (i.e., enzymatic lignin); however, most lignin is burned as low−calorific fuel to provide heat energy or directly discharged. Previous research showed that only approximately 2% of lignin could be found for high−value utilization [10,11]. In addition, the carbon content of lignin is up to 60%, which can make it a good precursor for preparing carbon materials. Therefore, the use of lignin as a precursor to prepare carbon materials has great application potential in the future [12,13,14], especially in the field of electrocatalysis. For example, Graglia [15] used beech lignin as a raw material to prepare microporous, mesoporous, and macroporous nitrogenous carbon materials through alkaline hydrothermal treatment, aromatic nitrification reaction, and roasting activation. Its E_1/2_ in alkaline medium was 0.85 V, showing ORR electrocatalytic activity comparable to that of the most advanced nonprecious metal catalysts. Zhang [16] prepared nitrogen and sulfur co−doped porous carbon with good electrical conductivity by roasting amine lignin sulfonate at high temperature, showing good ORR electrocatalytic activity. Li [17] first proposed synthesis of an Fe−N−C/FePx/NPSC catalyst with a high specific surface area and rich pore structure through a salt−assisted process using sodium lignosulfonate as a carbon precursor, which promoted the catalyst to expose more active sites, increasing the catalytic activity and stability. The half−wave potential E_1/2_ reached 0.9 V, which was superior to the half−wave potential (0.85 V) measured for commercial Pt/C.

It follows, then, that lignin is an excellent raw material for preparing electrocatalytic oxygen reduction catalysts. However, the sources and molecular structure of lignin are complex and variable, and the yield and repeatability of preparing lignin−based carbon materials are low, impeding the preparation and application of lignin−based carbon materials. Therefore, it is necessary to deeply study the basic chemical structure of lignin, and to develop a simple, green, low−cost, large−scale preparation method for the production of lignin−based carbon materials. In addition, alkaline lignin (AL) is the main byproduct in the process of pulp and papermaking and forest biomass refining, with an annual output of over 50 million tons in China. Enzymatic lignin (EL) is a new type of lignin that is separated from the residue of microbial enzymatic hydrolysis processes. Compared with traditional alkaline lignin and sulfate lignin, enzymatic lignin has a variety of active groups and better reactivity. Nevertheless, most research has focused on the utilization of AL in the field of electrocatalytic oxygen reduction [15,18,19,20,21], and little research has investigated the development and exploitation of EL. Whether EL is suitable for the preparation of electrocatalytic carbon materials, and what the differences are between lignins that come from different sources, remains to be verified. Therefore, the purpose of this paper is to identify the effects of specific lignin structures on the preparation of heteroatomic porous carbon’s electrocatalytic oxygen reduction performance, based on the differences in the structure and properties of different kinds of lignin, to broaden the application field of enzymatic hydrolysis of lignin. In this paper, N−doped porous carbon was prepared with enzymatic hydrolysis lignin (EL), alkaline lignin (AL) and dealkaline lignin (DL) as raw materials and melamine as a nitrogen source to explore the effects of the structural characteristics of different lignin raw materials on its properties.

## 2. Experimental Materials and Methods

### 2.1. Experimental Materials

EL was purchased from Shandong Longli Biotechnology Co. Ltd., dealkaline lignin (DL) and melamine (C_3_H_6_N_6_, 99%) were purchased from Macklin Chemical Reagent Co., Ltd. (Shanghai, China), AL was purchased from Aladdin Chemical Reagent Co., Ltd. (Shanghai, China), Nafion (Model D520 mass fraction 5%) was purchased from Shanghai Hesen Electric Co., Ltd., and ethanol (analytically pure, 99.7%) was purchased from Jiangsu Qiangsheng Functional Chemical Co., Ltd. (Changshu, China).

### 2.2. Experimental Procedure

First, 1 g of lignin (EL, AL, and DL) and 4 g of melamine were dispersed in 100 mL of deionized water (90 °C) and stirred for 30 min. Then, 1 g of ZnCl_2_ was added, heated in a water bath (100 °C), and the mixture was stirred until the deionized water evaporated completely, after which the residue was dried and ground. The sample was placed in a porcelain boat and annealed in a quartz tube. In a N_2_ atmosphere, the sample was heated to 900 °C at a rate of 5 °C min^−1^ (the N_2_ flow rate was 50~100 mL min^−1^), held for 2 h, and then naturally cooled to room temperature. The obtained lignin carbon material was pickled with 1 M hydrochloric acid and then dried to obtain a black lignin carbon material (LC). The mass ratio of lignin, ZnCl_2_, and melamine was 1:1:4, and the LC samples prepared using enzymatic hydrolysis lignin, alkaline lignin, and dealkaline lignin mixed with ZnCl_2_ and melamine were labeled N−ELC, N−ALC, and N−DLC, respectively.

### 2.3. Catalyst Structure Characterization

A Fourier−transform infrared spectrometer (FTIR, Germany, BRUKER−Vertex 70) and UV–Vis absorption spectra (UV−Vis, USA, PE−Lambda 750) were used to determine the types of functional groups in lignin molecules. The contents of different elements (C, H, N, S) in the materials were determined by using the element analyzer (Germany, Elementar−vario MACRO cube). The thermal stability of the materials was tested using an SDT Q600 analyzer (TA Instruments Inc., New Castle, DE, USA). X−ray diffraction (XRD; Germany, Brooker−Bruker D8 ADVANCE) was used to characterize the crystal structure of the materials. The surface composition of the samples was determined by X−ray photoelectron spectroscopy (XPS). Raman spectroscopy, using a Raman microscope, was used to characterize the defect degree of the materials. The morphologies of the carbon materials were characterized by high−resolution transmission electron microscopy (HRTEM) (Japan, JEOL−JEM 2100 F). The specific surface area and pore size distribution were determined by a nitrogen adsorption–desorption test (BET, USA, Mack−ASAP2460).

### 2.4. Electrocatalytic Performance Test of the Catalyst

The electrochemical tests were conducted using a Shanghai Chenhua CHI760E electrochemical workstation. In this experiment, a three−electrode system was used: the rotating disk electrode (RDE) was the working electrode, the saturated calomel electrode (SCE) was the reference electrode, the platinum plate electrode was the opposite electrode, and the electrolyte solution was 0.1 M KOH. The diameter of the glass carbon electrode of the RDE was 4 mm.

Catalyst samples (2 mg) were dispersed in a 550 μL mixed solution, including 250 μL of ethanol, 250 μL of deionized water, and 50 μL of Nafion solution (5 wt%), and ultrasonicated for 30 min to prepare ink−containing catalysts. The working electrode was prepared by absorbing 12 μL of ink, dropping it on the glass carbon electrode, and drying it naturally at room temperature. A comparison of commercial Pt/C (Johnson Matthey, 20 wt%) catalysts was used to prepare the ink and the working electrodes.

Cyclic voltammetry (CV) and linear sweep voltammetry (LSV) were performed at the working electrodes of the supported catalyst at 0.1 M KOH saturated with O_2_ and N_2_, respectively. The CV test parameters were in the potential scanning range of 0~1.2 V at a scanning rate of 50 mV·s^−1^. The LSV test parameters were in the potential scanning range of 0~1.2 V at a scanning rate of 5 mV·s^−1^ in an O_2_ atmosphere. Without special explanation, the polarization curve at 1600 rpm was selected for comparison in this paper.

## 3. Results and Discussion

### 3.1. Analysis of the Basic Properties of Different Lignins

#### 3.1.1. Element Analysis

The preparation process of carbon materials is a process in which carbon elements are rearranged and accompanied by other elements escaping under extreme temperature conditions. Therefore, it is very advantageous to analyze and explore the transformation process of the lignin structure during the preparation of carbon materials by analyzing the elemental composition of lignin molecules. Table 1 shows the elemental compositions of three kinds of lignin: EL, AL, and DL. As shown in Table 1, the C contents of EL, AL, and DL were above 65%, with little difference among them. The C content of AL was the highest (65.74%), while that of EL was lower (65.24%). The high carbon content of lignin proves that it is a valuable carbon source material. In addition, EL, AL, and DL all contain different proportions of trace N and S, which may be related to the different sources of lignin.

#### 3.1.2. Molecular Structure Analysis of Different Lignin Species

Figure 1 shows the FTIR spectra of the three different kinds of lignin. It can be seen from the figure that the absorption peaks of the different lignins are effectively the same, indicating the presence of similar functional groups in the molecular structures of several lignins. Several kinds of lignin exhibit strong broad peaks at 3406 cm^−1^, which can be attributed to the O−H stretching vibration absorption peaks of phenol and alcohol hydroxyl groups in lignin. The weaker absorption peaks at 2938 cm^−1^ and 2840 cm^−1^ represent C−H stretching vibrations in the methyl and methylene functional groups, respectively. The absorption peak in the range of 1602 cm^−1^~1425 cm^−1^ is caused by the stretching vibration of the aromatic skeleton [22]. The absorption peak at 1120 cm^−1^ is the characteristic absorption of the lilac−based structural unit, which exists in all three lignins. The absorption peak at 1029 cm^−1^ is the C−O bending vibration of secondary alcohol and ether bonds. The characteristic absorption peak at 832 cm^−1^ is the p−hydroxyphenyl (H)−type structural unit. It can be seen from the figure that there are p−hydroxyphenyl structural units in EL and DL, while no characteristic absorption peak of this structural unit can be seen in AL. The absorption peak of EL at 1700 cm^−1^ indicates that it has a more unconjugated carbonyl structure, which indicates that conjugated double bonds in EL are destroyed, and some oxidation or esterification reactions occur [23]. AL and DL exhibit absorption peaks at 620 cm^−1^, corresponding to aliphatic C−C stretching vibrations, while there is no absorption peak at this position in EL, which proves that EL does not contain this functional group.

Lignin molecules have ultraviolet absorption groups such as aromatic structures, C=C double bonds, and conjugated C=O groups, which can strongly absorb ultraviolet light. Therefore, ultraviolet absorption spectra can be used to analyze the characteristic peaks of lignin. As shown in Figure 2, EL, DL, and AL all have obvious aromatic ring characteristic absorption peaks at 296 nm, corresponding to the absorption caused by electron transition in conjugated molecular structures, such as aromatic rings. DL and AL showed wide absorption peaks near 340 nm, corresponding to absorption of coumaric acid and ferulic acid, indicating that these two lignins contained a certain amount of phenolic acid structures. Compared with DL and AL, there was no obvious absorption peak near 340 nm in EL, indicating that there were very few functional groups of phenolic acids in this lignin.

Figure 3a,b show the TG and DTG curves of three different kinds of lignin, respectively. According to the figure, the thermal decomposition process of EL can be divided into four stages: the first stage (20~150 °C) is mainly the evaporation of water in lignin, and the sample mass loss rate is 10.4%. The second stage occurs in the temperature range from 150 °C to 240 °C, which is mainly the stage of lignin depolymerization and the temperature range of lignin glass transition, and the sample mass loss rate is 15.8%. In the temperature range from 240 °C to 420 °C, the third stage is the main stage of lignin pyrolysis. During this heating process, lignin undergoes a thermal decomposition reaction, and the branched−chain structure of lignin is largely broken and transformed into a large number of volatile small molecular products (CO, CO_2_, H_2_O, etc.), with a mass loss rate of 49.4%. The DTG curve shows that the maximum weight loss rate peaks at approximately 333 °C. When the temperature is higher than 420 °C, the chemical bond around the benzene ring breaks, and the benzene ring also breaks and recombines to form aromatic compounds before polymerizing into amorphous carbon. At this time, the thermal decomposition process is essentially completed, which is the fourth stage of pyrolysis [24]. DL and EL have a similar thermal decomposition process, but the mass residual rate of DL is much higher than that of EL because more ash remains after the end of DL pyrolysis. For AL, the pyrolysis process is quite different. AL contains many phenolic hydroxyl groups, most of which are connected with other structural units in the form of ether bonds. The DTG curves show that two maximum values of the thermal decomposition rate appear at 237 °C and 420 °C. This pyrolysis process is mainly manifested as the general breaking of aryl ether bonds and methoxy groups, along with side−chain breaking through dehydration to produce small molecular compounds. The process of lignin carbonization is the rearrangement of carbon atoms at high temperature. Therefore, different thermal cracking processes may affect the structure of the final carbon material and, thus, further affect its electrocatalytic performance.

### 3.2. Effect of Catalyst Type and Amount on the Pyrolysis Product Distribution of Nitrogen−Doped Porous Carbons

Figure 4 shows TEM images of different lignin−based carbon materials. As shown in the figure, N−ALC presents the shape of nanoparticles (Figure 4b), while N−ELC and N−DLC show the morphology of carbon nanosheets, which are thin and transparent, showing a graphene−like transparent structure, and folds can be observed (Figure 4a,c). In the high−resolution TEM images (Figure 4d,f), the lattice spacing of the carbon material at approximately 0.34 nm can be clearly seen, which is attributed to the (002) crystal plane of graphitic carbon [19], while the lattice spacing at approximately 0.25 nm in Figure 4e is mainly attributed to the (101) graphitic carbon crystal plane.

It can be seen from the full XPS scan map in Figure 5a that the N 1_S_ peak is at 398.9 eV. The analysis of N types of porous carbon doped with different lignin−based N is shown in Figure 5b–d. The peaks at binding energies of approximately 397, 398, 399, and 400 eV correspond to pyridinic nitrogen, pyrrolic nitrogen, graphitic nitrogen, and nitrogen oxide, respectively. Table 2 shows the nitrogen contents and nitrogen types of different samples. It has been proven in many studies that the presence of pyridinic N can improve the onset potential of ORR, while graphitic N can improve the limiting diffusion current density [25]. The highest content of graphitic nitrogen in N−ELC was 43%, and that of pyridinic nitrogen was 25.1%, both of which were higher than those in N−ALC and N−DLC. Therefore, the ORR catalytic performance of N−ELC should theoretically be superior to that of N−ALC and N−DLC. However, the electrocatalytic performance is also affected by the specific surface area and pore volume [26].

Raman characterization can be used to further analyze the changes in the defect degree of the carbon material structure. As shown in Figure 6a, the D band (1350 cm^−1^) and G band (1580 cm^−1^) represent the lattice defects of carbon atoms and the stretching vibration of the sp^2^ hybrid orbital plane of the carbon atoms, respectively, while the D−band–G−band ratio (I_D_/I_G_) reflects the LC defect degree of the carbon materials. The smaller the I_D_/I_G_, the higher the graphitization degree and crystallinity of the sample. The I_D_/I_G_ ratios of N−ELC, N−ALC, and N−DLC were 1.21, 1.01, and 1.33, respectively. The I_D_/I_G_ values of N−ALC were the lowest, and the graphitization degree was relatively high among several lignins. The maximum I_D_/I_G_ value of N−DLC indicates that there are more defects in this material [19].

At the same time, the crystal structures of different lignin carbon materials were characterized by XRD. The XRD patterns of N−ELC, N−ALC, and N−DLC showed two wide diffraction peaks (Figure 6b), whose centers were at approximately 25° and 43°, corresponding to the (002) and (101) crystal planes of graphitized carbon, respectively. Therefore, by using the strategy of melamine−assisted co−catalysis, the three lignin raw materials can produce relatively good graphite−structured carbon nanomaterials, which is consistent with the TEM results.

Specific surface area is a key factor affecting the oxygen reduction activity of carbon materials. A high specific surface area is conducive to exposing more oxygen reduction active sites, thereby improving the catalytic performance of the ORR [27]. Figure 7a shows the N_2_ adsorption and desorption curves of different carbon materials. N−ELC, N−ALC, and N−DLC all show type−IV isotherm characteristics in the medium–high−pressure region (P/P_0_ = 0.45~1), and there are obvious retention rings, indicating that there are abundant mesoporous structures. The pore size distribution was determined using the Barrett−Joyner−Halenda (BJH) equation, as shown in Figure 7b. The average pore size of the three carbon materials was mostly between 2~50 nm, among which N−ALC had an abundant mesoporous structure. The BET specific surface areas of N−ELC, N−ALC, and N−DLC were 55.82, 332.83, and 60.02 m^2^·g^−1^, respectively. The abundant mesoporous structure and large specific surface area of N−ALC may increase the exposed active sites and promote the mass diffusion and electron transfer of electrocatalysis, thereby achieving better electrocatalytic performance.

### 3.3. Electrocatalytic Oxygen Reduction Performance of Different Kinds of Lignin Carbon Materials

The ORR catalytic performance of lignin carbon materials was determined by cyclic voltammetry (CV). Cyclic voltammetry curves of the materials were measured in 0.1 M KOH solutions saturated with O_2_ and N_2_ (Figure 8a–c). It can be seen from the test results that the CV curves of the three different lignin–carbon catalysts in the N_2_−saturated electrolyte do not show obvious redox peaks. In the O_2_−saturated electrolyte, obvious reduction peaks appeared at approximately 0.82, 0.84, and 0.67 V, indicating that these carbon materials have electrocatalytic ORR activity.

Figure 9a shows the LSV curves of different lignin carbon materials at 1600 rpm, while Figure 9b shows a comparison of important visual indicators of electrocatalytic performance: half−wave potential (E_1/2_), onset potential (E_onset_), and limiting current density (j_d_). As shown in Figure 9, N−DLC has the worst catalytic performance among the three lignins, while N−ALC and N−ELC both have good ORR performance. The E_1/2_ and E_onset_ of N−ALC are 0.84 and 0.96 V, respectively, while the E_1/2_ and E_onset_ of N−ELC are 0.82 and 0.93 V, respectively. Compared with the standard commercial performance of 20% Pt/C (E_1/2_ = 0.86 V, E_onset_ = 0.98 V), the catalytic performance of N−ELC reached more than 95% of its catalytic performance. This indicated that compared with common alkaline lignin, although the specific surface area and pore volume of EL were smaller than those of AL, the ORR catalytic performance of EL was slightly lower than that of AL, but both had excellent catalytic performance. This indicates that EL could also be used as an excellent electrocatalyst raw material. The electrochemical performance parameters of biomass−based oxygen reduction electrocatalysts are listed in Appendix A (Appendix A).

## 4. Conclusions

In this paper, different lignin–nitrogen−doped porous carbon catalysts were prepared using EL, AL, and DL as carbon sources and melamine as the nitrogen source. The results show that, like AL, EL is also an excellent carbon source material for preparing electrocatalysts. The FTIR and UV−Vis results showed that the surface functional groups of the three kinds of lignin were similar. The thermal performance analysis showed that EL and AL had different pyrolysis processes, and that of AL was mainly a process of aryl ether bond breaking, methoxy group breaking, and side−chain dehydration forming small molecules. The electrocatalytic oxygen reduction results showed that N−ALC had the best ORR performance among the three materials. The E_1/2_ and E_onset_ of N−ALC were 0.84 V and 0.96 V, respectively. The E_1/2_ and E_onset_ of N−ELC were 0.82 and 0.93 V, respectively—slightly lower than those of N−ALC. Compared with the standard commercial performance of 20% Pt/C (E_1/2_ = 0.86 V, E_onset_ = 0.98 V), the catalytic performance of N−ELC can reach more than 95% of its catalytic performance.

## Figures and Tables

**Figure 1 ijerph-20-04383-f001:**
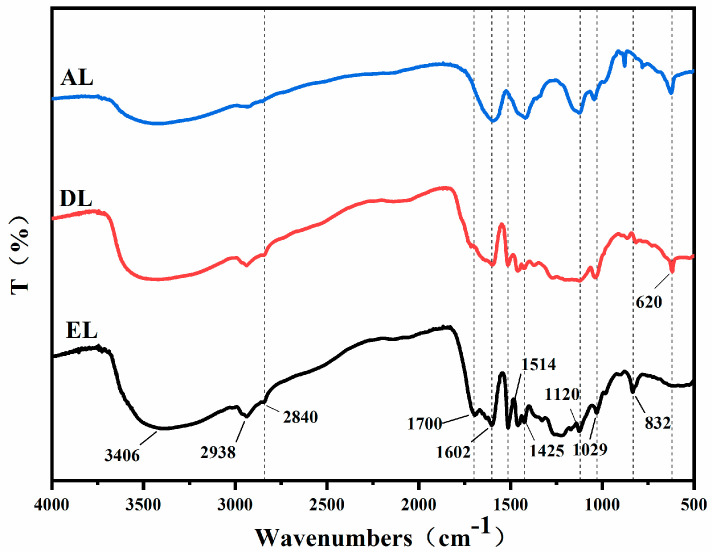
FTIR of different kinds of lignin.

**Figure 2 ijerph-20-04383-f002:**
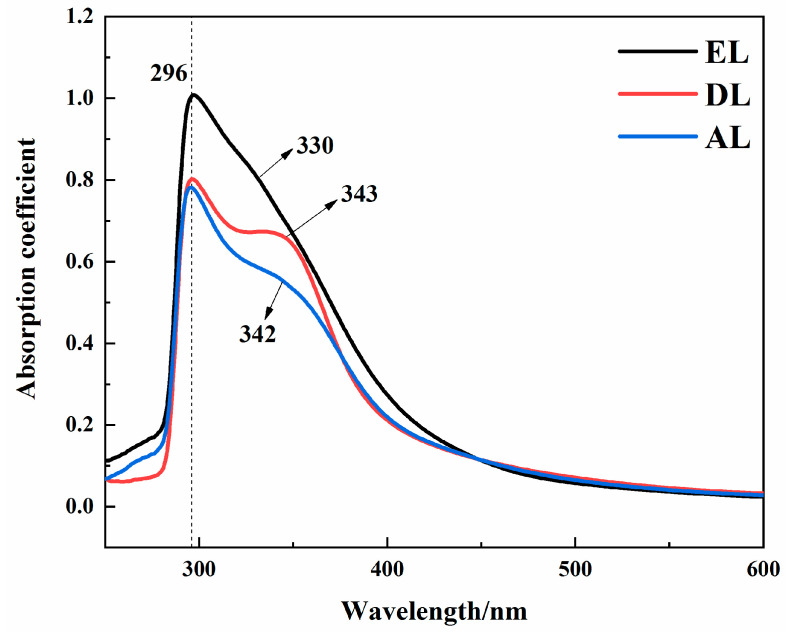
UV−Vis spectra of different kinds of lignin.

**Figure 3 ijerph-20-04383-f003:**
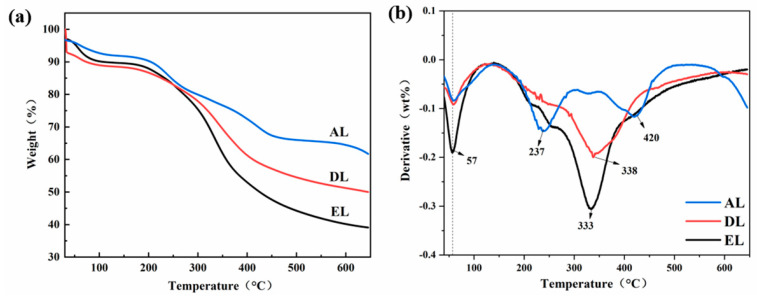
Thermal performance analysis of different kinds of lignin: (**a**) TG curve; (**b**) DTG curve.

**Figure 4 ijerph-20-04383-f004:**
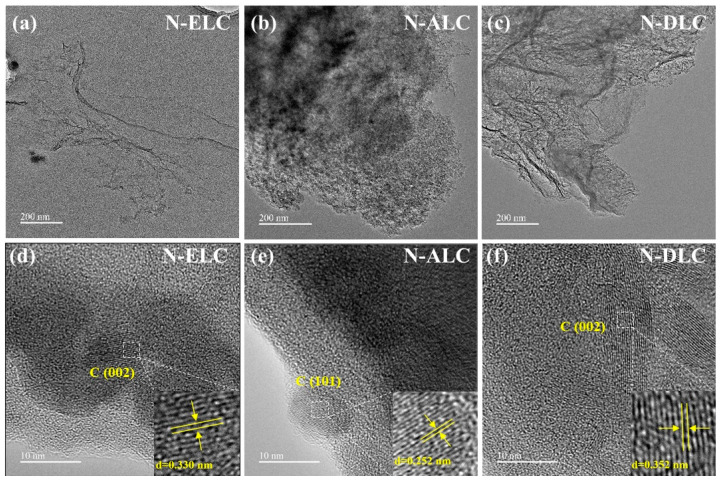
TEM diagrams of different kinds of lignin carbon materials: (**a**,**d**) N−ELC; (**b**,**e**) N−ALC; (**c**,**f**) N−DLC.

**Figure 5 ijerph-20-04383-f005:**
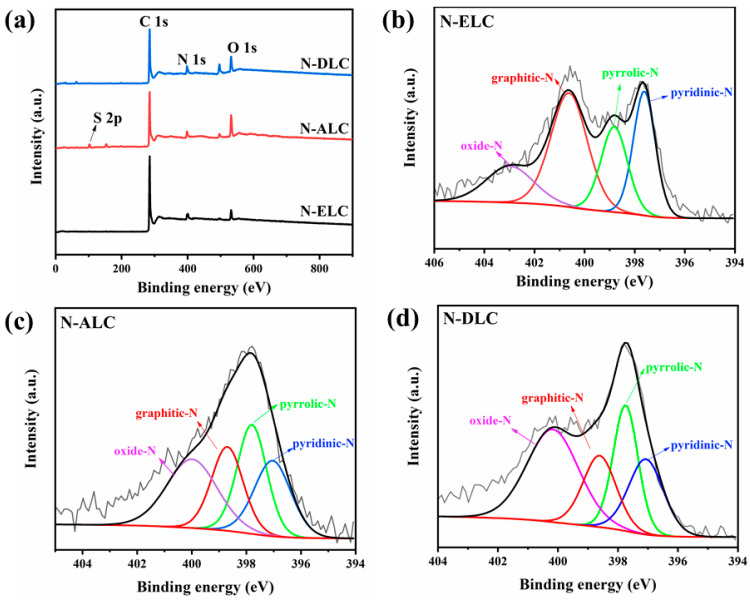
(**a**) XPS full−scan spectra of different lignin carbon materials, and N 1s spectra of (**b**) N−ELC, (**c**) N−ALC, and (**d**) N−DLC.

**Figure 6 ijerph-20-04383-f006:**
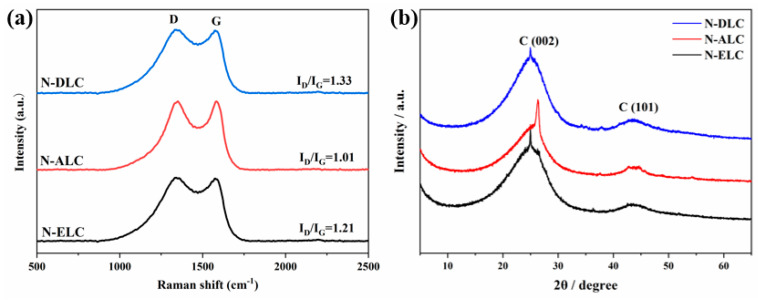
(**a**) Raman spectra and (**b**) XRD spectra of different kinds of lignin carbon materials.

**Figure 7 ijerph-20-04383-f007:**
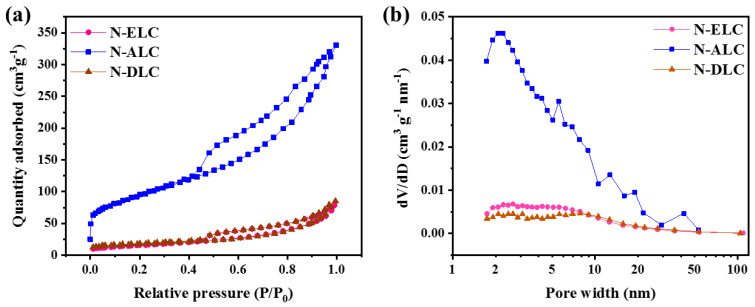
(**a**) N_2_ adsorption–desorption isotherms of different lignin carbon materials. (**b**) Aperture distribution curve.

**Figure 8 ijerph-20-04383-f008:**
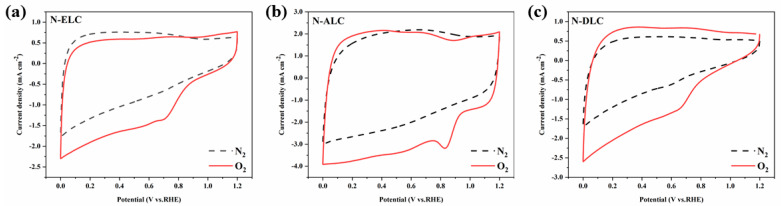
CV curves of different catalysts in O_2_− and N_2_−saturated electrolytes. (**a**) N−ELC; (**b**) N−ALC; (**c**) N−DLC.

**Figure 9 ijerph-20-04383-f009:**
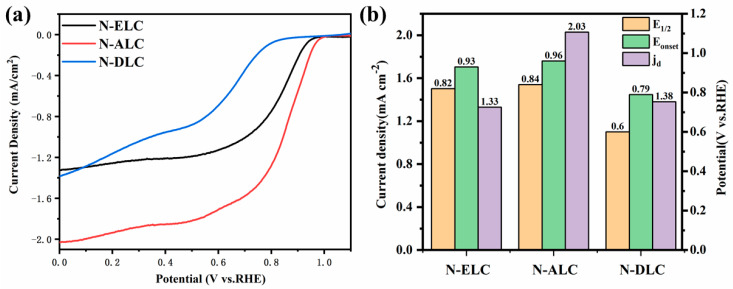
ORR performance: (**a**) LSV curve obtained at 1600 rpm. (**b**) Performance comparison bar chart.

**Table 1 ijerph-20-04383-t001:** Chemical characteristics of the nitrogen−doped porous carbons.

Lignin Sources	Element Content (%)			
	C	H	N	S
EL	65.24	5.38	1.05	0.16
AL	65.74	7.30	0.84	0.71
DL	65.45	5.26	0.31	2.56

**Table 2 ijerph-20-04383-t002:** N contents and N types of different lignin materials.

Lignin Carbon Materials	N Content (%)	Percentage of Functional Group Area (%)			
		N Oxide	Graphitic N	Pyrrolic N	Pyridinic N
N−ELC	8.89	12.7	43.0	19.3	25.1
N−ALC	7.33	28.1	22.4	26.3	23.2
N−DLC	11.38	38.0	17.9	24.1	20.0

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
