# Peer review of "Impact of Different Lignin Sources on Nitrogen−Doped Porous Carbon toward the Electrocatalytic Oxygen Reduction Reaction"

_ijerph, 2023, doi:10.3390/ijerph20054383_

Round 1

Reviewer 1 Report

The article entitled "Impact of different lignin sources on nitrogen-doped porous carbon toward the electrocatalytic oxygen reduction reaction" deals with electrocatalysts derived from different types of lignin. The authors obtained three types of lignin-based catalysts and thoroughly characterized them. It was found that the N-ALC showed the best electrocatalytic performance among other materials. The results are reliable and the methods are sufficient. I recommend to publish this article after minor checks.

1. The description of FTIR, elemental analysis, UV-spectroscopy, TGA should be added into the experimental part. 

2. Will the activity of the lignin-derived catalyst be increased if metals (e.g. Fe) are supported? Other words, is further modification of material needed?

Author Response

Dear Reviewers,

On behalf of my co-authors, we are very grateful to you for giving us an opportunity to revise our manuscript. We appreciate you very much for your positive and constructive comments and suggestions on our manuscript entitled “Impact of different lignin sources on nitrogen-doped porous carbon toward the electrocatalytic oxygen reduction reaction” (Manuscript ID: ijerph-2192950).

   We have studied comments carefully and tried to our best to revise our manuscript according to the comments. The following are the response and revisions I have made in response to your questions and suggestions on an item-by-item basis. The modifications are marked in red in the manuscript. Thanks again to the hard work of the editor and reviewer!

Reviewer 2 Report

The authors synthesized lignin-based carbon materials and studied the effects of various lignin sources on the performance of electrocatalytic oxygen reduction of porous carbon catalysts. The manuscript is well written. The manuscript can be accepted after minor corrections.

1. Authors should compare their results with the reported ones.

2. Why N-ALC shows distant behavior in BET analysis as compared with N-ELC and N-DLC samples?

3. In XRD, why was peak shifting observed for the sample N-ALC?

4. Few topological and grammatical mistakes should be resolved. 

Author Response

(The authors gave the same response as above.)

Reviewer 3 Report

Please see attached PDF.

Author Response

(The authors gave the same response as above.)
